# Investigating the effect of nickel concentration on phytoplankton growth to assess potential side-effects of ocean alkalinity enhancement

Jiaying Abby Guo[1], Robert Strzepek[2], Anusuya Willis[3], Aaron Ferderer[1], Lennart Thomas Bach[1]

1 Institute for Marine and Antarctic Studies, University of Tasmania, Hobart, Tasmania, Australia

2 Australian Antarctic Program Partnership (AAPP), Institute for Marine and Antarctic Studies, University of Tasmania, Hobart, Tasmania, Australia

3 National Collections and Marine Infrastructure, Commonwealth Scientific and Industrial Research Organisation, Hobart, Tasmania, Australia

*Correspondence to*:    Jiaying Abby Guo (jiaying.guo@utas.edu.au)

**Abstract.** Ocean alkalinity enhancement (OAE) is a proposed method for removing carbon dioxide ($CO_2$) from the atmosphere by the accelerated weathering of (ultra-) basic minerals to increase alkalinity – the chemical capacity of seawater to store $CO_2$. During the weathering of OAE-relevant minerals relatively large amounts of trace metals will be released and may perturb pelagic ecosystems. Nickel (Ni) is of particular concern as it is abundant in olivine, one of the most widely considered minerals for OAE. However, so far there is limited knowledge about the impact of Ni on marine biota including phytoplankton. To fill this knowledge gap, this study tested the growth and photo-physiological response of 11 marine phytoplankton species to a wide range of dissolved Ni concentrations (from 0.07 nmol/L to 50,000 nmol/L). We found that the phytoplankton species were not very sensitive to Ni concentrations under the culturing conditions established in our experiments, but the responses were species-specific. The growth rates of 6 of the 11 tested species showed generally limited but still significant responses to changing Ni concentrations (36% maximum change). Photosynthetic performance, assessed by measuring the maximum quantum yield ($F_v/F_m$) and the functional absorption cross-section ($\sigma_{PSII}$) of photosystem II, was sensitive to changing Ni in 3 out of 11 species (35% maximum change) and 4 out of 11 species (16% maximum change), respectively. The limited effect of Ni may be partly due to the provision of nitrate as the nitrogen source for growth, as previous studies suggest higher sensitivities when urea is the nitrogen source. Furthermore, the limited influence may be due to the relatively high concentrations of synthetic organic ligands added the growth media in our experiments. These ligands are commonly added to control trace metal

bioavailability and therefore for example "free $Ni^{2+}$" concentrations by binding the majority of the dissolved Ni. Our data suggest that dissolved Ni does not have a strong effect on phytoplankton under our experimental conditions, but we emphasize that a deeper understanding of nitrogen sources, ligand concentrations and phytoplankton composition is needed when assessing the influence of Ni release associated with OAE.

## 1 Introduction

Increased burning of fossil fuels and land-use changes have resulted in a significant increase in atmospheric $CO_2$ from a preindustrial value of ~280 ppm to currently ~415 ppm (Friedlingstein et al., 2020). Detrimental effects of rising $CO_2$ include global warming, increasing sea levels, ocean acidification and more frequent extreme weather (IPCC, 2019). To limit detrimental impacts, $CO_2$ emissions must be rapidly reduced. Additionally, about 100-1000 gigatonnes (Gt) of $CO_2$ must be removed from the atmosphere by 2100 and permanently stored in other reservoirs (Rogelj et al., 2018). One potential method for the required atmospheric $CO_2$ removal (CDR) is to increase ocean alkalinity thereby increasing the chemical capacity of seawater to permanently store $CO_2$ (Kheshgi, 1995). Alkalinity is formed naturally during the chemical weathering of certain minerals rich in magnesium or calcium such as olivine (Schuiling and Krijgsman, 2006). When these minerals are dissolved in the ocean, protons are consumed reducing seawater $CO_2$ concentrations thereby causing an enhanced $CO_2$ influx from the atmosphere or a reduced flux to the atmosphere.

Natural rock weathering will absorb most of the anthropogenic $CO_2$, but only over a period of tens- to hundreds-thousand years (Archer et al., 2009). "Ocean alkalinity enhancement (OAE)" and "enhanced weathering (EW)" seek to accelerate natural rock weathering processes by spreading pulverized minerals onto the ocean surface (in the case of OAE) or warm and humid land areas (in the case of EW) (Schuiling and Krijgsman, 2006; Kheshgi, 1995). Modeling studies suggest that OAE and EW can help to mitigate climate change significantly when operated at an appropriate scale (Lenton et al., 2018; Ilyina et al., 2013; Kohler et al., 2010; Keller et al., 2014).

A variety of trace metals are released into the environment alongside alkalinity during chemical weathering. The composition and quantity of released trace metals depends on the mineral used for OAE or EW. Olivine is currently one of the most widely considered minerals due to its relatively fast weathering rates (Taylor et al., 2016; Oelkers et al., 2018). It contains high amounts of nickel (Ni), which was shown to leach out of olivine very efficiently during chemical weathering (Montserrat et al., 2017; Fuhr et al., 2022). Thus, the potentially large amounts of Ni

released into the environment are a predominant environmental concern of EW or OAE with olivine (Hartmann et al., 2013; Bach et al., 2019). In the case of EW, Ni would first affect terrestrial ecosystems but a fraction of it would be transported into the oceans via rivers. In the case of OAE, Ni would directly affect marine biota. Phytoplankton are at the base of the marine food web so that it is central to the assessment of EW and OAE to

understand how phytoplankton species respond to Ni perturbations (Bach et al., 2019).

Dissolved Ni occurs in low concentrations (2-4 nmol/L) in the sea surface, but concentrations increase with depth (up to 11 nmol/L) in the North Pacific, the Atlantic, and the Indian Ocean (Bruland, 1980; Sclater et al., 1976; Middag, 2020; Thi Dieu Vu and Sohrin, 2013). The depletion in the surface in some ocean regions is thought to be caused by phytoplankton utilization of dissolved Ni and the enrichment with depth due to remineralization of

exported particulate Ni (Glass and Dupont, 2017; Dupont et al., 2010; Morel, 2008). The nutrient-like vertical profile of Ni indicates that it is a bioactive element for phytoplankton in some areas (Glass and Dupont, 2017). Indeed, Ni is an essential co-factor for some enzymes (Zamble, et al. 2017; Sunda, 1989) and two major functions of Ni for phytoplankton metabolism have been documented. First, Ni is known to be involved in urea utilization. Urea is an ecologically important nitrogen source that can support 5-50% of oceanic primary production (Wafar et

al., 1995). Most marine phytoplankton, including cyanobacteria, haptophytes, dinoflagellates, and diatoms, use the Ni-containing enzyme urease to hydrolyse urea to ammonium and $CO_2$ ($(NH_2)_2CO + H_2O \rightarrow CO_2 + 2NH_3$) (Holm and Sander, 1997; Dupont et al., 2010). Second, Ni can be a co-factor for the enzyme superoxide dismutase (SOD) (Wolfe-Simon et al., 2005). SOD is important for the survival of photosynthetic organisms (Glass and Dupont, 2017). The highly reactive and noxious superoxide anion radical ($O_2^-$) is a metabolic by-product of aerobic

respiration and oxygenic photosynthesis (Fridovich, 1998). SOD can turn $O_2^-$ into molecular oxygen ($O_2$) and hydrogen peroxide ($H_2O_2$). For $N_2$-fixers, nitrogenase is a key enzyme for dinitrogen ($N_2$) fixation. Since nitrogenase can be inactivated by reactive oxygen species, such as $O_2^-$, Ni-SOD is indirectly involved in nitrogen fixation process in cyanobacteria. In addition, hydrogen ($H_2$) is generated as a by-product in the nitrogen fixation process, and Ni is an essential part of the hydrogenase enzymes regulating $H_2$ metabolism used by some $N_2$-fixers

(Tuo, et al. 2020). Hence, Ni plays a role in cyanobacterial $N_2$ fixation in different ways.

This project tested the response of 11 different marine phytoplankton species to a gradient of dissolved Ni concentrations. The phytoplankton species were exposed to this gradient under the same experimental conditions. We address the following questions: (1) how do different dissolved Ni concentrations influence phytoplankton growth and photosynthetic performance? (2) will different phytoplankton species or functional groups have

different Ni sensitivities?

**2 Materials and Methods**

Eleven axenic cultures from four different phytoplankton functional groups (diatoms, haptophytes, cyanobacteria, and dinoflagellates) were obtained from the Australian National Algae Culture Collection. We selected species from temperate regions as they can be grown at the same temperature and seawater medium. Selected species

included three diatoms: *Asterionellopsis glacialis* (CS-135), *Nitzschia closterium* (CS-5), *Phaeodactylum tricornutum* (CS-29); four haptophytes: *Cricosphaera* sp. (CS-1183), *Emiliania huxleyi* (CS-1185), *Isochrysis galbana* (CS-186), *Prymnesium parvum* (CS-659); three cyanobacteria: *Geitlerinema* sp. (CS-897), *Oscillatoria* sp. (CS-52), *Synechococcus* sp. (CS-205, sub-cluster 5.2 and pigment type 1 (only phycocyanin)); and one dinoflagellate *Amphidinium carterae* (CS-740).

**2.1 Growing phytoplankton in artificial seawater medium**

This study used Aquil medium due to its wide application in trace metal experiments (Price et al., 1989). The medium is composed of artificial seawater where Milli-Q 18.2 MΩ·cm$^{-1}$ grade water is mixed with ultra-pure salts to reproduce the major ion composition of seawater (Pausch et al., 2019). The medium was filtered through a 0.2 μm pore size filter and sterilized in a microwave for a total of 11 minutes in acid-cleaned polycarbonate bottles (2

L) (Price et al., 1989). This artificial seawater is further enriched with the elements necessary for algal growth, such as vitamins, macronutrients (nitrate ($NO_3^-$) = 100 μmol/L, phosphate ($PO_4^{3-}$) = 10 μmol/L, and silicate ($SiO_3^{2-}$) = 100 μmol/L), and various essential trace metals such as iron and manganese (Table A1). The trace metals were buffered with 100 μmol/L ethylenediaminetetraacetic acid (EDTA). Seventeen Aquil media were produced that differed in the amount of Ni that was added, as will be described in more detail in the next section. Media

preparation was done in a trace metal clean laminar flow bench. The salinity and pH (NBS scale) of Aquil media were 35 and 8.1 respectively.

Phytoplankton species were cultivated in acid-cleaned (10% HCl for at least 24 h) polycarbonate tubes (30 mL, Nalgene $^{TM}$). These polycarbonate tubes (one tray with 40 tubes) were filled with Milli-Q water and then sterilized in the microwave for 8 minutes. The Aquil media were transferred from 1 L bottles into empty polycarbonate tubes

under the clean bench under the trace metal clean laminar flow bench. Phytoplankton were added to the medium once it had reached chemical equilibrium (see next section).

The cultures were grown in a light chamber at 17°C. All polycarbonate tubes were mounted onto a self-made "phytoplankton disc", which rotated at 0.8 revolutions per minute (Fig. 1 (a)). The phytoplankton disc ensured that equal light intensity was provided to all cultures and that phytoplankton cells were kept in suspension. The light was provided on a 14-to-10-hour daily cycle (cool white fluorescence light) where light intensities were 58 μmol photons $m^{-2}$ $s^{-1}$ (14 hours) and 23 μmol photons $m^{-2}$ $s^{-1}$ (10 hours). This unusual light cycle was due to some lights in the room being plugged in energy sources, which had separate light-dark cycle setups linked to the computer system at the Institute of Marine and Antarctic Studies. Initially, we were not aware of this additional cycle and only realized the issue during the experiment. Therefore, we continued with this light cycle to maintain comparability between experiments. However, this issue does not affect the interpretation of the results as all species and replicates received the same amount of light throughout the experiment. The light intensity was the average light intensity at each of the 88 spots on the phytoplankton disc measured with a Licor light meter.

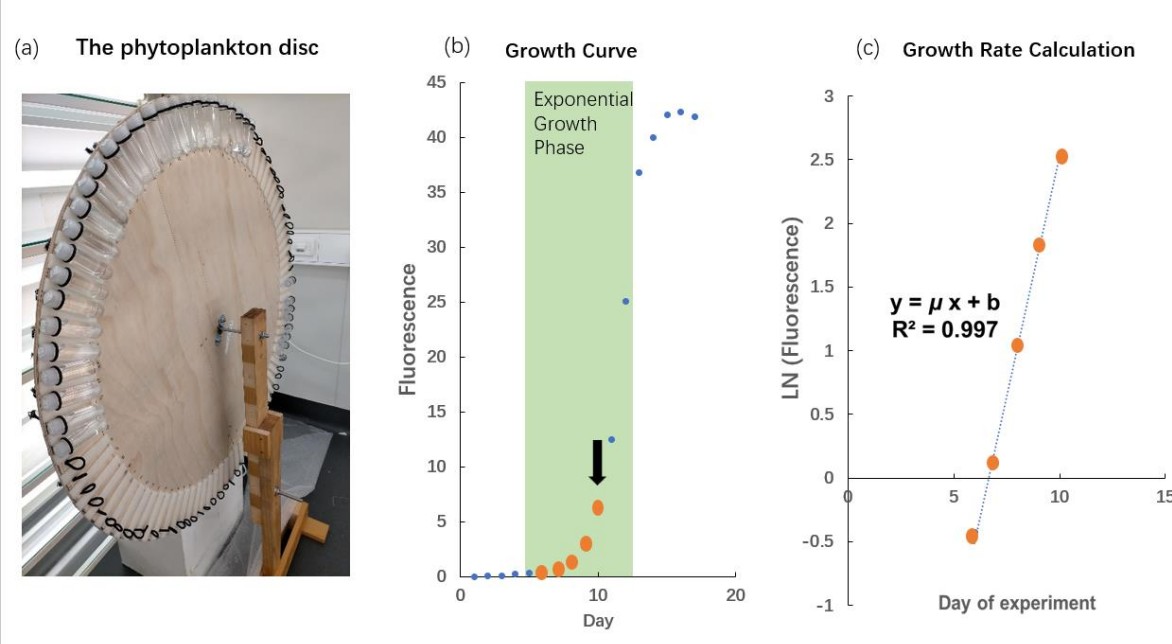

**Figure 1.** The phytoplankton disc and growth rate calculation. (a) The phytoplankton disc, with polycarbonate tubes mounted using elastic bands to the edge of the circular disc. The disc rotated with 0.8 revolutions per minute during the experiment. (b) In vivo chlorophyll fluorescence during the growth cycle of phytoplankton cultures, *Phaeodactylum tricornutum* (CS-29). We only used fluorescence values where biomass inside the polycarbonate tubes was still relatively low (maximum up to fluorescence of 6.1) as indicated in this example with the thick orange dots. The arrow indicates the time when the culture was usually transferred into the next batch of fresh medium. (Please note that the data illustrated here is from a test where we let the culture grow into nutrient depletion.) (c) The fluorescence values measured at low biomass were ln-transformed and plotted against time (day). The slope of the linear regression in this plot represents the specific growth rate ($\mu$; $d^{-1}$).

**2.2 Nickel treatment**

Aquil media were enriched with different concentrations of $NiCl_2$: 0, 5, 10, 20, 30, 50, 70, 100, 150, 200, 300, 400, 500, 700, 1000, 10000 and 50000 nmol/L. Unless otherwise noted, "Ni concentration" refers to the total added dissolved Ni concentration. For illustration and discussion of the data, concentrations were negatively $\log_{10}$ transformed:

$$pNi = -\log_{10}(Ni) \hspace{4cm} Eq. (1)$$

where Ni is the total dissolved concentration of Ni in mol/L. This kind of transformation is also used to convert hydrogen ion concentrations to pH and is commonly used in studies investigating trace metal sensitivities to better visualize data when trace metal concentrations vary over orders of magnitude (Dupont et al., 2008).

Media were allowed to equilibrate chemically for at least 24 h before being inoculated with phytoplankton. To acclimate the phytoplankton strains, stock cultures were first transferred into Aquil medium without Ni enrichment. They were then cultivated for at least 3 batch cycles (i.e., transferred from one polycarbonate tube to the next one) before being transferred to polycarbonate tubes with the different Ni treatments. This ensured that the phytoplankton species were acclimated to Aquil medium before the Ni experiment commenced.

EDTA binds with metal ions and helps the dissolution of metal ions to create a nutrient-replete medium. Due to the addition of the ligand EDTA to the Aquil media, the "free Ni" ion concentrations (i.e., $Ni^{2+}$) were substantially lower than the total dissolved Ni concentrations as was calculated with the chemical speciation software Visual MINTEQ 3.1 (Gustafsson, 2011).

We were interested to see if the response of phytoplankton to Ni may be different in other growth media where no EDTA was added. Therefore, we prepared a batch of natural seawater medium with water sampled from 15 m in the Southern Ocean (58.02 °S, 141.17 °E). There was little information about concentrations and types of Ni-binding organic ligands in the Southern Ocean because these ligands occur at very low concentrations within a highly complex mixture of organic matter (Boiteau et al., 2016). If we take Fe-binding organic ligands as examples: the characterized types of Fe-binding organic ligands were different in various studies due to the diverse measuring protocol, and the concentrations of these ligands in the Southern Ocean varied from 0.72 to 12.3 nmol/L (Nolting et al., 1998; Boye et al. 2001; Buck et al., 2010). Therefore, the Southern Ocean seawater we used in the experiment can be considered to have much lower organic ligands than the Aquil media (100 µmol/L EDTA). This natural seawater was filtered through an acid-cleaned 0.2 µm filter and sterilized in the microwave. The same amount of macro-nutrients (N, P and Si) and vitamins were added as in the Aquil medium (mentioned above). The trace metal

additions to the Southern Ocean seawater (no Ni included) were adjusted to a similar free trace metal concentration (nutrient-replete) as in Aquil medium (Table A1). For the experiment with natural seawater, we set up a dissolved Ni gradient with 17 concentrations: 0, 1, 2, 5, 10, 20, 30, 50, 70, 100, 150, 200, 300, 400, 500, 700, and 1000 nmol/L. The extremely high Ni concentrations designed for the Aquil medium were avoided as we assumed the organic ligand concentrations in natural seawater to be much lower than the concentration of EDTA added in Aquil medium and therefore the concentration of free $Ni^{2+}$ to be higher. We used *P. tricornutum* (CS-29) for this experiment. *Phaeodactylum tricornutum* was transferred from the stock cultures into natural seawater medium for 3 batches cycles prior to the experiment with different Ni treatments as described for the Aquil medium above.

The total ion concentrations of each trace metal in natural seawater and Aquil media before additions were measured using a seaFAST system and inductively-coupled plasma mass spectrometry (ICP-MS). The free ion concentrations were calculated with Visual MINTEQ 3.1 based on the total ion concentration together with the added concentration (Table 1).

**Table 1.** The total dissolved concentrations and free ion concentrations of Ni in different media.

| Aquil medium | | | | | Southern Ocean seawater medium | | | | |
|---|---|---|---|---|---|---|---|---|---|
| Added Ni concentration (nmol/L) | Total dissolved Ni concentration (mol/L) | pNi | Free $Ni^{2+}$ concentration (mol/L) | $pNi^{2+}$ | Added Ni concentration (nmol/L) | Total dissolved Ni concentration (mol/L) | pNi | Free $Ni^{2+}$ concentration (mol/L) | $pNi^{2+}$ |
| 0 | $7.1 \times 10^{-11}$ | 10.2 | $9.4 \times 10^{-17}$ | 16.0 | 0 | $8.6 \times 10^{-9}$ | 8.1 | $6.1 \times 10^{-9}$ | 8.2 |
| 5 | $5.1 \times 10^{-9}$ | 8.3 | $6.8 \times 10^{-15}$ | 14.2 | 1 | $9.6 \times 10^{-9}$ | 8.0 | $6.8 \times 10^{-9}$ | 8.2 |
| 10 | $1.0 \times 10^{-8}$ | 8.0 | $1.3 \times 10^{-14}$ | 13.9 | 2 | $1.1 \times 10^{-8}$ | 8.0 | $7.5 \times 10^{-9}$ | 8.1 |
| 20 | $2.0 \times 10^{-8}$ | 7.7 | $2.7 \times 10^{-14}$ | 13.6 | 5 | $1.4 \times 10^{-8}$ | 7.9 | $9.7 \times 10^{-9}$ | 8.0 |
| 30 | $3.0 \times 10^{-8}$ | 7.5 | $4.0 \times 10^{-14}$ | 13.4 | 10 | $1.9 \times 10^{-8}$ | 7.7 | $1.3 \times 10^{-8}$ | 7.9 |
| 50 | $5.0 \times 10^{-8}$ | 7.3 | $6.7 \times 10^{-14}$ | 13.2 | 20 | $2.9 \times 10^{-8}$ | 7.5 | $2.0 \times 10^{-8}$ | 7.7 |
| 70 | $7.0 \times 10^{-8}$ | 7.2 | $9.3 \times 10^{-14}$ | 13.0 | 30 | $3.9 \times 10^{-8}$ | 7.4 | $2.7 \times 10^{-8}$ | 7.6 |
| 100 | $1.0 \times 10^{-7}$ | 7.0 | $1.3 \times 10^{-13}$ | 12.9 | 50 | $5.9 \times 10^{-8}$ | 7.2 | $4.2 \times 10^{-8}$ | 7.4 |
| 150 | $1.5 \times 10^{-7}$ | 6.8 | $2.0 \times 10^{-13}$ | 12.7 | 70 | $7.9 \times 10^{-8}$ | 7.1 | $5.6 \times 10^{-8}$ | 7.3 |
| 200 | $2.0 \times 10^{-7}$ | 6.7 | $2.7 \times 10^{-13}$ | 12.6 | 100 | $1.1 \times 10^{-7}$ | 7.0 | $7.8 \times 10^{-8}$ | 7.1 |
| 300 | $3.0 \times 10^{-7}$ | 6.5 | $4.0 \times 10^{-13}$ | 12.4 | 150 | $1.6 \times 10^{-7}$ | 6.8 | $1.1 \times 10^{-7}$ | 7.0 |
| 400 | $4.0 \times 10^{-7}$ | 6.4 | $5.4 \times 10^{-13}$ | 12.3 | 200 | $2.1 \times 10^{-7}$ | 6.7 | $1.5 \times 10^{-7}$ | 6.8 |
| 500 | $5.0 \times 10^{-7}$ | 6.3 | $6.7 \times 10^{-13}$ | 12.2 | 300 | $3.1 \times 10^{-7}$ | 6.5 | $2.2 \times 10^{-7}$ | 6.7 |
| 700 | $7.0 \times 10^{-7}$ | 6.2 | $9.4 \times 10^{-13}$ | 12.0 | 400 | $4.1 \times 10^{-7}$ | 6.4 | $2.9 \times 10^{-7}$ | 6.5 |
| 1000 | $1.0 \times 10^{-6}$ | 6.0 | $1.4 \times 10^{-12}$ | 11.9 | 500 | $5.1 \times 10^{-7}$ | 6.3 | $3.6 \times 10^{-7}$ | 6.4 |
| 10000 | $1.0 \times 10^{-5}$ | 5.0 | $1.5 \times 10^{-11}$ | 10.8 | 700 | $7.1 \times 10^{-7}$ | 6.2 | $5.0 \times 10^{-7}$ | 6.3 |
| 50000 | $5.0 \times 10^{-5}$ | 4.3 | $1.4 \times 10^{-10}$ | 9.9 | 1000 | $1.0 \times 10^{-6}$ | 6.0 | $7.2 \times 10^{-7}$ | 6.1 |

## 2.3 Growth rate measurement

Growth rate measurements were conducted according to the methods described by Andersen (2005). Briefly, the chlorophyll fluorescence of the cells was recorded daily at the same time of the day with a Turner Model 10-AU fluorometer. During the measurements, polycarbonate tubes did not have to be opened because they fit inside the sample chamber of the fluorometer. This reduced the risk of contamination as the polycarbonate tubes remained closed throughout the batch cycles. Fluorescence signals of samples were measured after 20 minutes of dark acclimation. The fluorescence values were *ln*-transformed and plotted as a function of incubation days. A linear regression was fitted during the exponential phase of phytoplankton growth with the specific growth rate ($\mu$; d$^{-1}$) represented by the slope of the linear regression (Fig. 1(b) and (c)). We only used fluorescence values up to 13 (arbitrary unit) for our growth rate calculations so that the biomass in the incubation bottles remained relatively low and consistent with the dilute batch culture principle (Laroche et al., 2010).

Reliable estimates of exponential growth rates in dilute batch cultures require multiple serial transfers of cultures (all performed while the strain is still in exponential growth) to allow the time for cultures to acclimate to the experimental conditions (Brand et al., 1981; Andersen, 2005). Therefore, the phytoplankton species were transferred into new polycarbonate tubes containing fresh medium during their early exponential stage for 3 batch cycles prior to recording growth rates shown in the results. This meant that cultures were usually growing in their respective treatment conditions for at least three weeks.

## 2.4 Fast repetition rate fluorometry

We conducted photo-physiological measurements at the end of each batch cycle. A Fast repetition rate (FRR) fluorometry (Fast Ocean Sensor FRRf3, Chelsea Instruments Group) was used to measure the maximum quantum yield, $F_v/F_m$, and the functional absorption cross-section of photosystem II ($\sigma_{PSII}$; nm$^2$ reaction centre (RC)$^{-1}$). These measurements were done with cultures directly after they had been used to inoculate the subsequent batch cycle (hence avoiding contamination of ongoing cultures). Cultures were kept in dark for 20 minutes before the measurements. For each treatment and species, 5ml phytoplankton samples were added to the FRR fluorometry cuvette, which was temperature-controlled at 17 °C. Filtered Aquil media (or natural seawater media) were used at the beginning of the measurement for blank calibration. Throughout the experiment, FRR fluorometry was used with an acquisition sequence of 100 saturation flashes for 200 μs, 40 relaxation flashes for 2.4 ms, while the flash

duration was set to 100 μs (Schallenberg et al., 2020). In each acquisition sequence, three channels with different light wavelengths were used: channel A with 450 nm light; channel B with 450 nm and 530 nm light; and channel C with 450 nm and 624 nm. The FRR fluorescence results from channel A (450 nm) were used to analyse diatoms, haptophytes, and dinoflagellates photosynthetic performance due to the presence of chlorophyll a in their cells, while channel C (450 nm and 624 nm) results were used to analyse the photosynthetic performance of cyanobacteria because of the presence of the phycobilin which is commonly present in cyanobacteria (Roy et al., 2011). At least 10 acquisitions were measured for each sample and used to calculate the average value of $F_v/F_m$ and $\sigma_{PSII}$. $F_v/F_m$ is usually lower under nutrient or light stress (summarized by Suggett et al. (2009)), while $\sigma_{PSII}$ describes the ability of light to promote a photochemical reaction in PSII (Falkowski and Raven, 1997). The value of $F_v/F_m$ and $\sigma_{PSII}$ are known to vary among algal taxa (Suggett et al., 2009). Typically, cells growing in batch cultures at the exponential growth phase exhibit a constant value of $F_v/F_m$ and $\sigma_{PSII}$ (Parkhill et al., 2001).

**2.5 Data analysis**

Every strain was able to grow in all Ni concentrations in Aquil media for at least 3 batch cycles. The data from the third batch were used for analyses. The growth rate and photo-physiological response of phytoplankton was analysed using generalised additive models (GAMs) and plotted in RStudio (R packages "mgcv" and "ggplot2") (RStudio team, 2020). For the GAM analyses, we assumed that growth rates, $F_v/F_m$ and $\sigma_{PSII}$ of phytoplankton would show an optimum curve in response to the wide range of Ni concentrations: Ni limitation at the lower extremes, Ni inhibition at the upper extremes and an optimum at some intermediate Ni concentration. GAMs were fitted to plots to assess the presence of a relationship between Ni concentration, growth rates and photo-physiological responses. P-values of the smooth terms of GAM models greater than 0.05 indicated that there was no statistically significant trend in the response variable ($\mu$, $F_v/F_m$ or $\sigma_{PSII}$) in response to the wide Ni gradient (i.e., the smooth term was not significantly different from a horizontal line and therefore no statistically significant relationship between Ni and the measured parameter present). The general GAM equation is:

$$Y = I_0 + S(pNi) + e \qquad\qquad \text{Eq. (2)}$$

where $Y$ is the response variable ($\mu$, $F_v/F_m$ and $\sigma_{PSII}$); $I_0$ is the intercept; $S(pNi)$ is the non-parametric smooth function according to pNi; and $e$ the error. The k-value (basis dimension) of GAM formula in Rstudio was set to the minimum k-value that fitted the curve and explained the data points without fitting random noise. The function 'gam.check' in the package "mgcv" was used to assess the appropriateness of the selected k-value following Wood

 (2022). The selection of k involved ensuring the p-value was greater than 0.05. This ensured the selected k-value

was sufficiently small to not fit random noise (i.e overfitting) but still statistically appropriate.

## 3 Results

### 3.1 Growth rates comparison

235 Most trace metals in seawater are partially bound by organic ligands and their bioavailable "free" concentrations

are lower than the total dissolved ion concentrations (Van Den Berg and Nimmo, 1987). The thermodynamic

equilibrium concentrations of the $-\log_{10}$ transformed "free Ni concentrations" ($pNi^{2+}$) and total dissolved Ni

concentrations (pNi) in the different media (see Eq. 1 and Table 1.) correlate linearly ($R^2 > 0.99$). Thus, both can

be displayed as separate x-axes on the same plot (Fig. 2, 3 and 4). For the Southern Ocean seawater media we

240 assumed the ligand concentration to be 0 and thus that the differences between pNi and $pNi^{2+}$ are very small. In

Aquil, however, the differences between pNi and $pNi^{2+}$ are very large due to the presence of EDTA.

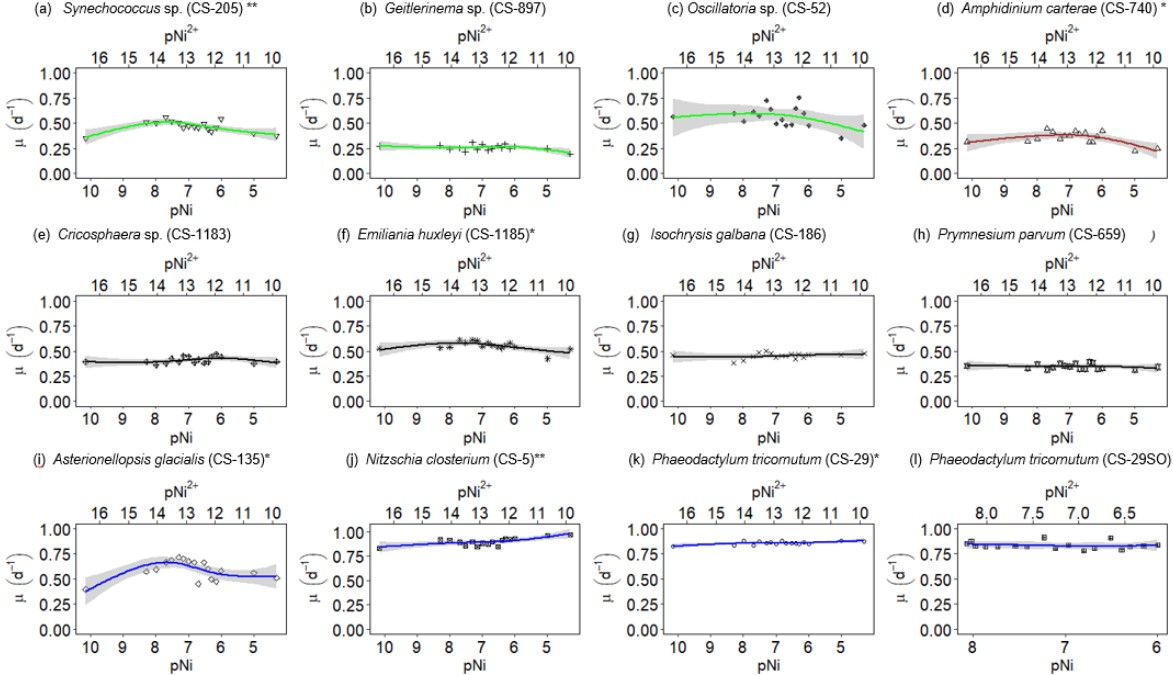

**Figure 2.** Growth rates of different phytoplankton strains in a large gradient of Ni concentrations. The species name is shown
245 in each subplot with the strain number in the parentheses. pNi and $pNi^{2+}$ are the $-\log_{10}$ transformed values of the total
dissolved and free Ni concentrations, respectively (Eq. (1)). A smaller value represents a higher concentration. Plots (a)-(c)
are cyanobacteria; plot (d) is a dinoflagellate; plots (e)-(h) are haptophytes; plots (i)-(l) are diatoms. Plots (a)-(k) show

growth rates in Aquil media while plot (l) shows growth rates of *P. tricornutum* in natural seawater media. Solid lines represent the smooth terms produced from GAMs using the growth rate data and pNi concentrations. Shading indicates the 95% confidence interval. P-values <0.05 indicate that the smooth term is significantly different from a straight horizontal line. P-values <0.05 are indicated by an *, and P-values <0.01 are indicated by ** after the species names.

Six out of the 11 strains displayed statistically significant growth rate changes in response to Ni sensitivity (Fig. 2, Table 2). These strains were *Synechococcus* sp. (CS-205), *A. carterae* (CS-740), *E. huxleyi* (CS-1185), *A. glacialis* (CS-135), *N. closterium* (CS-5), and *P. tricornutum* (CS-29). Among these strains, *N. closterium* (CS-5) and *P. tricornutum* (CS-29) had consistent increasing growth rates when pNi increased (Fig. 2 (j) and (k)). Other strains displayed optimum curves response patterns, although variations in growth rates between the low, high, and optimum concentration of pNi and these trends were below 36% (Table A3). Most of their optimal growth rates were in the range of pNi 8-7 (10 nmol/L to 100 nmol/L). Growth rates of the other strains (*Geitlerinema* sp. (CS-897), *Oscillatoria* sp. (CS-52), *Cricosphaera* sp. (CS-1183), *I. galbana* (CS-186), and *P. parvum* (CS-659)) were not significantly affected by different Ni concentrations.

**Table 2.** Approximate significance of the smooth terms for GAMs. Three separate GAMs were used to calculate the impacts of pNi on $\mu$, $F_v/F_m$ and $\sigma_{PSII}$ (Eq. 2). P-values <0.05 indicate that the smooth term is significantly different from a straight line. P-values <0.05 are indicated by an *, and P-values <0.01 are indicated by **. Adj r² is the adjusted r squared value. DE stands for deviance explained.

| Strain number | Strain name | P-value of $\mu$ | | Adj r² | DE | P-value of $F_v/F_m$ | | Adj r² | DE | P-value of $\sigma_{PSII}$ | | Adj r² | DE |
|---|---|---|---|---|---|---|---|---|---|---|---|---|---|
| CS-205 | *Synechococcus* sp. | 0.005 | ** | 0.554 | 0.628 | 0.003 | ** | 0.515 | 0.574 | 0.508 | | -0.035 | 0.030 |
| CS-897 | *Geitlerinema* sp. | 0.313 | | 0.086 | 0.168 | 0.003 | ** | 0.528 | 0.585 | 0.173 | | 0.156 | 0.235 |
| CS-52 | *Oscillatoria* sp. | 0.282 | | 0.099 | 0.184 | 0.114 | | 0.102 | 0.158 | 0.068 | | 0.262 | 0.347 |
| CS-740 | *Amphidinium carterae* | 0.019 | * | 0.433 | 0.518 | 0.030 | * | 0.228 | 0.276 | <0.001 | ** | 0.783 | 0.822 |
| CS-1183 | *Cricosphaera* sp. | 0.504 | | -0.034 | 0.030 | 0.574 | | -0.044 | 0.022 | 0.287 | | 0.104 | 0.198 |
| CS-1185 | *Emiliania huxleyi* | 0.048 | * | 0.342 | 0.434 | 0.588 | | -0.045 | 0.020 | 0.178 | | 0.251 | 0.380 |
| CS-186 | *Isochrysis galbana* | 0.347 | | -0.004 | 0.059 | 0.697 | | -0.056 | 0.010 | 0.763 | | -0.026 | 0.058 |
| CS-659 | *Prymnesium parvum* | 0.510 | | -0.035 | 0.030 | 0.348 | | -0.004 | 0.059 | 0.003 | ** | 0.519 | 0.577 |
| CS-135 | *Asterionellopsis glacialis* | 0.034 | * | 0.418 | 0.512 | 0.080 | | 0.246 | 0.332 | 0.006 | ** | 0.528 | 0.601 |
| CS-5 | *Nitzschia closterium* | 0.004 | ** | 0.459 | 0.500 | 0.120 | | 0.097 | 0.153 | 0.328 | | 0.001 | 0.064 |
| CS-29 | *Phaeodactylum tricornutum* | 0.013 | * | 0.300 | 0.344 | 0.389 | | 0.066 | 0.157 | 0.025 | * | 0.356 | 0.432 |

| | | | | | | | | |
|---|---|---|---|---|---|---|---|---|
| CS-29SO | *Phaeodactylum tricornutum* | 0.517 | -0.036 | 0.029 | 0.662 | -0.018 | 0.062 | 0.260 | 0.117 | 0.208 |

The cyanobacterium *Oscillatoria* sp. (CS-52) tended to aggregate during culturing and the fluorescence signals were more variable on a day-to-day basis. This made the growth rate calculation less accurate, indicated by lower
$R^2$ values in linear regression when fitting ln-transformed data over time to calculate the growth rate.

We were interested if we could trust singular datapoints at the extreme ends of the optimum curves, as they often drove trends in our data (e.g., *Synechococcus* in Fig. 2 at pNi <7.5, total dissolved Ni < 30 nmol/L). Therefore, we did an additional experiment with *Synechococcus* sp. (CS-205) where we replicated the lowest added Ni treatment (0 nmol/L; 0.07 nmol/L including background Ni) and the optimum Ni concentration (20 nmol/L) (Table 3). The
results confirmed the trend in the optimum curve, with the added 20 nmol/L Ni resulting in significantly enhanced growth rates (Table 3).

**Table 3.** Physiological responses of *Synechococcus* sp. (CS-205) at two different Ni concentrations with 3 replicates each treatment (shown individually). Ni con. is the total dissolved Ni concentration in the media. $\mu$ means growth rate (day$^{-1}$). SD
means standard deviation. P-value was calculated using T-test. The unit of $\sigma_{PSII}$ is nm$^2$ reaction centre (RC)$^{-1}$.

| Ni con. (nmol/L) | $\mu$ | Average $\mu$ | SD | P-value of $\mu$ | Fv/Fm | Average Fv/Fm | SD | P-value of Fv/Fm | $\sigma_{PSII}$ | Average $\sigma_{PSII}$ | SD | P-value of $\sigma_{PSII}$ |
|---|---|---|---|---|---|---|---|---|---|---|---|---|
| 0 | 0.25 | | | | 0.33 | | | | 228 | | | |
| 0 | 0.31 | 0.30 | 0.04 | | 0.34 | 0.35 | 0.02 | | 236 | 230 | 6.23 | |
| 0 | 0.33 | | | 0.001 | 0.37 | | | 0.179 | 224 | | | 0.797 |
| 20 | 0.54 | | | | 0.41 | | | | 227 | | 21.6 | |
| 20 | 0.51 | 0.52 | 0.03 | | 0.41 | 0.38 | 0.04 | | 247 | 226 | | |
| 20 | 0.49 | | | | 0.34 | | | | 204 | | | |

### 3.2 Photosynthesis performance of phytoplankton

The FRR fluorescence data were largely consistent with the growth rate data in that no strong trends within the Ni range tested were observed for most of species. The $\sigma_{PSII}$ and $F_v/F_m$ measurements across the Ni gradient revealed
minimal trends, with generally little variation between treatments (Fig. 3 and 4). A few exceptions to this general pattern results are mentioned below.

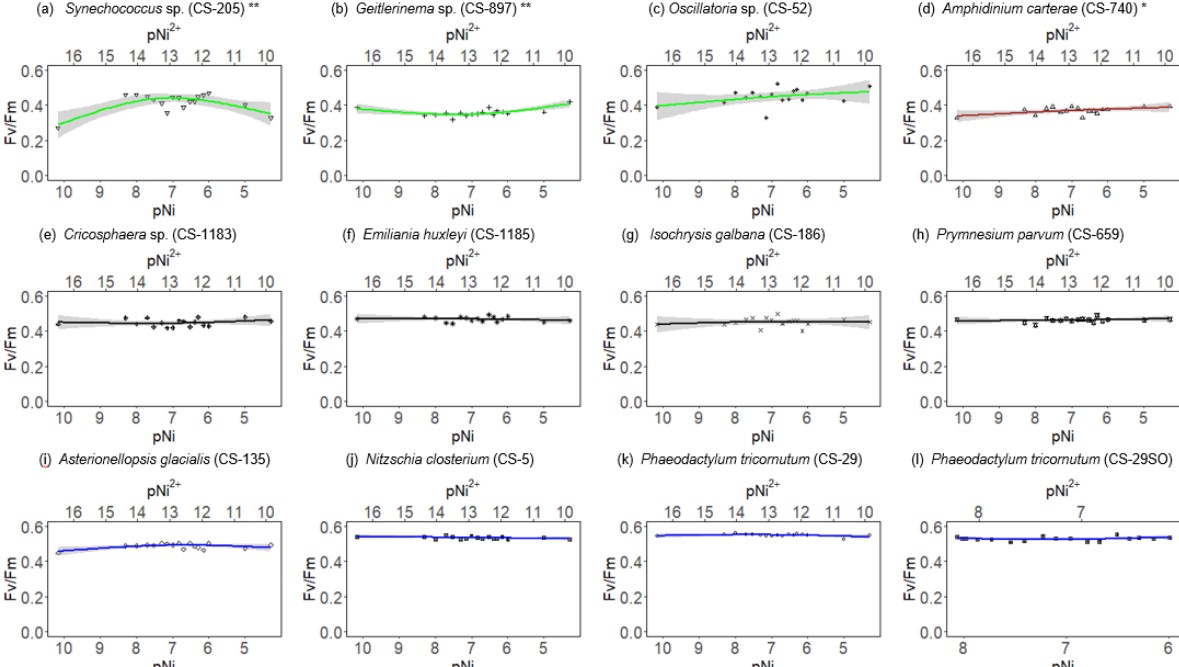

**Figure 3.** $F_v/F_m$ results of phytoplankton cultures. pNi and pNi$^{2+}$ are the -log$_{10}$ transformed values of the total dissolved and free Ni$^{2+}$ concentrations (Eq. (1)). A smaller value represents a higher concentration. Plots (a)-(c) are cyanobacteria; the plot (d) is a dinoflagellate; plots (e)-(h) are haptophytes; plots (i)-(l) are diatoms. Plots (a)-(k) were from strains growing in Aquil media while plot (l) are results for *P. tricornutum* growing in natural seawater media. Solid lines represent the smooth terms produced from GAMs using the growth rate data and pNi concentrations. Shading indicates the 95% confidence interval. P-values <0.05, indicate that the smooth term is significantly different from a straight line. P-values <0.05 are indicated by an *, and P-values <0.01 are indicated by **.

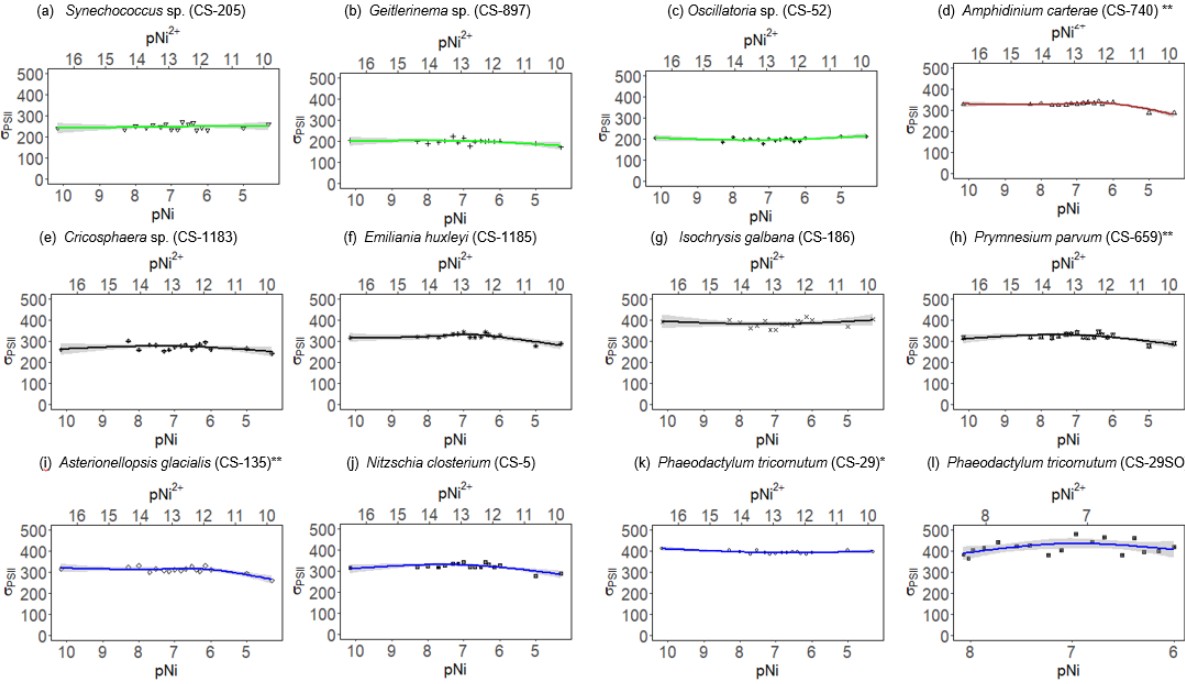

**Figure 4.** σ$_{PSII}$ results of phytoplankton cultures. pNi and pNi$^{2+}$ are the -log$_{10}$ transformed values of the total dissolved and free

Ni$^{2+}$ concentrations (Eq. (1)). A smaller value represents a higher concentration. The unit of $\sigma_{PSII}$ is nm$^2$ reaction centre (RC)$^{-1}$. Plots (a)-(c) are cyanobacteria; the plot (d) is a dinoflagellate; plots (e)-(h) are haptophytes; plots (i)-(l) are diatoms. Plots (a)-(k) were from strains growing in Aquil media while plot (l) are results for *P. tricornutum* growing in natural seawater media. Solid lines represent the smooth terms produced from GAM models using the growth rate data and pNi concentrations. Shading indicates the 95% confidence interval. P-values <0.05, indicate that the smooth term is significantly different from a straight

line. P-values <0.05 are indicated by an *, and P-values <0.01 are indicated by **.

*Synechococcus* sp. (CS-205) had higher $F_v/F_m$ values in the mid-pNi range (pNi 8-6, 10-1000 nmol/L) and the lowest $F_v/F_m$ value was in the Aquil medium without any Ni addition. In contrast, *Geitlerinema* sp. (CS-897) had lower $F_v/F_m$ values in the mid-pNi range but the variation between maximum and minimum $F_v/F_m$ values was

small. These two strains, however, exhibited little change in $\sigma_{PSII}$ over the range of Ni treatment. Some species (e.g., *A. carterae* (CS-740), *P. parvum* (CS-659) and *A. glacialis* (CS-135)) had slightly lower $\sigma_{PSII}$ values at the highest Ni concentrations (>10000 nM) suggesting some reduction in light harvesting capacity at high Ni concentrations. The small P-value of the smooth terms (Table 2) are likely driven by these low $\sigma_{PSII}$ values in the high Ni concentrations. In general, most of the tested strains appeared photosynthetically healthy across the tested

Ni gradient.

The most pronounced effect of Ni was observed in $F_v/F_m$ for *Synechococcus* sp. (Fig. 3). $F_v/F_m$ was considerably lower at pNi=10.2 (0.07 nmol/L) than at the optimum concentrations (approximately pNi=7.7, 20 nmol/L). Our additional experiment with *Synechococcus*, where we replicated the pNi 10.2 and 7.7 treatments three times, did not confirm this trend (Table 3). Neither $F_v/F_m$ nor $\sigma_{PSII}$ values were significantly different between the two Ni

concentrations (Table 3).

### 3.3 Comparison between Aquil media and the natural seawater media

*Phaeodactylum tricornutum* (CS-29) growing in the natural Southern Ocean seawater media (see 2.2) showed no significant trend ($F_v/F_m$) or particularly strong changes (growth rate, $\sigma_{PSII}$) across the experimental Ni concentration gradient. This result was very similar to the result of *P. tricornutum* (CS-29) grown in Aquil media. The average

growth rate of *P. tricornutum* growing in the Southern Ocean seawater media was 0.83 d$^{-1}$, which was very similar to the growth rates of the cultures growing in Aquil media (0.86 d$^{-1}$) (Fig. 2 (k)). Absolute numbers were also very similar for the $\sigma_{PSII}$ and $F_v/F_m$ data (Fig. 3 and 4).

**4 Discussion**

**4.1 Phytoplankton sensitivities to different Ni concentrations**

Based on growth rates and FRR fluorescence results, we conclude that changes in dissolved Ni, within the range tested and under the experimental conditions, do not have a strong effect on the 11 phytoplankton species. Only 4 species showed significant trends in both growth rates and at least one photophysiological parameter. Eight out of 11 species showed <25% and <16% change relative to the average values in growth rates and photo-physiological parameters, respectively (Table A3.). An exception was *Synechococcus* sp. (CS-205), which showed a significant

and quite pronounced growth rate enhancement of 74% from the lowest to optimum Ni (20 nmol/L) and then gradually declining growth rates towards the highest Ni. Likewise, growth rates of *A. carterae* (CS-740) showed a relatively pronounced Ni-sensitivity, following an optimum curve with highest growth rates between a pNi of 8-7 (10-100 nmol/L). The Ni sensitivity of growth rates in the other species where significant trends were detected were smaller, i.e. smaller than 25% change relative to the average growth rate of the species (Table A3). However,

we emphasise that even a small difference in growth rate can have a pronounced effect on population sizes during extended periods of growth due to the exponential nature of phytoplankton reproduction. For example, an increase of growth rate by 0.05 d$^{-1}$ (as frequently observed in our data; Fig. 2) would lead to a ~65% larger population at the end of a 10-day growth period. Furthermore, even if a species is completely insensitive to Ni it may still be affected indirectly within a competitive environment with multiple phytoplankton species present. This is because

other species may benefit from, or be inhibited by, changing Ni concentration thereby altering the competition for nutrient resources. Therefore, small changes in growth rates should not be readily marginalised as they may still be of ecological and biogeochemical relevance.

The inhibition of growth rate or photosynthesis performance was evident in a few species when pNi reached 5 (10000 nmol/L) (i.e., Fig. 2(a) and (d)), but most species did not have growth inhibition in high Ni concentration.

The relatively small effects of high Ni on growth rates, $F_v/F_m$ and $\sigma_{PSII}$ were surprising because we expected stronger species-dependent Ni sensitivity within the pNi range of 9-5, at least based on the available experimental evidence summarized by Glass and Dupont (2017). There are several potential reasons for the disagreement on the Ni sensitivity results from previous research. These will be discussed in the following subsections.

### 4.1.1 Dependency of Ni sensitivity on nitrogen sources

It has been reported that phytoplankton species have different Ni sensitivities depending on the nitrogen (N) source supporting growth. Oliveira and Antia (1986) found that 9 out of 12 phytoplankton species tested in their experiments showed faster growth when urea-enriched growth medium was supplemented with Ni. In contrast, no or less benefit of Ni was observed when the same species were grown in nitrate-enriched medium. Very similar observations of a growth-enhancing effect of Ni only when urea is the N source were later made by Price and Morel

(1991) and Egleston and Morel (2008) in experiments with two diatom (*Thalassiosira*) species. Based on these previous findings we conclude that the generally limited sensitivities observed in our study are partially due to the chosen N source.

In the oceans, nitrate fuels large parts of new primary production, i.e., production based on allochthonous nitrogen inputs to the euphotic zone (Eppley and Peterson, 1979). For example, nitrate is a key N source for new primary

production in upwelling regions such as the Southern Ocean (Maccready and Quay, 2001) and Eastern Boundary Upwelling Systems (Messié et al., 2009). It is also mixed into the surface during winter mixing and therefore important for new production during the phytoplankton spring bloom in temperate regions (Sieracki et al., 1993). Although the role of urea in marine primary production is less studied than the role of nitrate and ammonium, it likely plays an important role (Wafar et al., 1995). Like nitrate, urea can also be of allochthonous origin and

therefore by definition support new primary production. This is likely to occur in coastal regions where urea runoff from land, amplified by sewage effluents and agricultural activities, can be significant (Glibert et al., 2006). Urea is also produced during heterotrophic mineralization (Glibert et al., 2006), therefore constituting a predominant source for regenerated primary production – i.e., production based on remineralized nutrient sources (Eppley and Peterson, 1979). Indeed, ship-board enrichment experiments in the North Pacific have shown that urea strongly

enhances phytoplankton growth, especially the growth of the cyanobacterium *Prochlorococcus* (Shilova et al., 2017). We therefore conclude that the wide-spread relevance of urea for phytoplankton growth, and the dependence of urea cycling on Ni, suggests that Ni-sensitivities of many phytoplankton species may be more pronounced in real-world conditions than our simplified laboratory experiments would suggest.

### 4.1.2 Dependency of Ni sensitivity on organic ligand concentration

Organic ligands can chelate dissolved trace metals thereby changing their chemical speciation (Van Den Berg and

Nimmo, 1987). It is currently not known what chemical species of dissolved Ni influence phytoplankton physiology. If phytoplankton can access the total dissolved Ni pool, then experiments with different ligand concentrations could be more easily compared. However, most research suggests that phytoplankton are not primarily sensitive to the total dissolved Ni concentration but interact with free $Ni^{2+}$ ions (Dupont et al., 2010; Hudson and Morel, 1993; Morel et al., 1991). Free $Ni^{2+}$ only constitutes a fraction of the total dissolved Ni concentration depending on the organic ligand concentration (Donat et al., 1994). Unfortunately, ligands are chemically diverse and difficult to measure, meaning that their influence may not always be accounted for, and comparability between studies is difficult.

Our culture medium (Aquil) contained 100 μmol/L EDTA. Thus, despite dissolved Ni concentrations up to 50 μmol/L, free $Ni^{2+}$ concentrations were maximally 0.14 nmol/L. These concentrations will be lower than in other studies where comparable amounts of total Ni, but less EDTA was added. In natural seawater, organic ligands concentrations vary widely between regions. In regions with relatively high ligand concentrations, such as freshwater, 99.9% of dissolved Ni can be complexed (Xue et al., 2001). In seawater, generally 10%-50% of the total dissolved Ni is complexed by ligands depending on the region (Achterberg and Van Den Berg, 1997; Donat et al., 1994; Byrne, 2003; Saito et al., 2004).

The current understanding of organic complexation of Ni in surface seawater suggests that free $Ni^{2+}$ ion concentrations are generally not orders of magnitude lower than total dissolved Ni. For example, considering that most surface seawater has a total Ni concentration of 2-10 nmol/L, free $Ni^{2+}$ should approximately be within ~1-9 nmol/L based on the 10%-50% complexation in seawater mentioned above. These free $Ni^{2+}$ concentrations are considerably higher than the highest free $Ni^{2+}$ in our experiments (0.14 nmol/L), which raises the question if our experimental setup was suitable to test the influence of high Ni on phytoplankton. Answering this question is difficult as it is uncertain if total dissolved Ni concentrations influence phytoplankton physiology or only free $Ni^{2+}$ does. Our observation of decreasing growth rates in some of the phytoplankton species in the high Ni concentrations may be seen as a hint that dissolved Ni concentrations do play a role as it seems unlikely that the marginal increases in free $Ni^{2+}$ would induce Ni inhibition. Likewise, the almost identical growth and photo-physiological responses to Ni of *Phaeodactylum* grown in Aquil (with high EDTA ligand concentration) and natural Southern Ocean seawater (with presumably much lower ligand concentrations) could suggest that not only free $Ni^{2+}$ is important. Either way, these observations underscore the importance of organic ligands when studying Ni sensitivity of phytoplankton.

### 4.1.3 Species-specific Ni sensitivity due to enzyme requirements

Our results are consistent with earlier studies showing that different phytoplankton species have different Ni-sensitivities (e.g., Glass and Dupont, 2017; Oliveira and Anitia, 1986; Dupont et al., 2008). Species-specific sensitivities can be due to the different role of Ni as a co-factor for the enzyme SOD, which catalyses the conversion of $O_2^-$ to $O_2$ and $H_2O_2$. There are different kinds of SODs, with differing trace metal co-factor requirements. Typically, cyanobacteria utilize either Ni-SOD alone or combinations of manganese (Mn-) and Ni-SOD or iron (Fe-) and Mn-SOD. Diatoms and rhodophytes retain an active Mn-SOD, whereas chlorophytes, haptophytes, and embryophytes have either Fe-SOD or multiple combinations of Fe, Mn, and copper-zinc SODs (Wolfe-Simon et al., 2005). Ho (2013) has shown that Ni depletion limits Ni-SOD synthesis and nitrogen fixation rates in *Trichodesmium*. Moreover, Ni-SOD may be involved in the protection of the nitrogenase enzyme from superoxide inhibition during photosynthesis (Ho, 2013). Compared with other phytoplankton functional groups, cyanobacteria seem to rely more than other species on Ni-SOD which may explain their relatively high Ni-sensitivity (Dupont et al., 2008; Ho, 2013).

### 4.2 Implications for the assessment of ocean alkalinity enhancement

OAE can be achieved by distributing pulverized rocks on land and ocean surfaces, thereby accelerating chemical weathering rates and the generation of alkalinity. The environmental perturbation depends on the chemical composition of the applied rock minerals. If dunite is used as the source rock for OAE (an olivine-rich ultrabasic rock, often associated with vulcanism), the Ni perturbation could be particularly high as dissolution experiments with olivine powder found roughly a 3 μmol/kg increase in dissolved Ni for a ~100 μmol/kg increase in alkalinity within approximately 50 days (Montserrat et al., 2017). However, it is difficult to estimate how the free $Ni^{2+}$ concentration will change, because it depends on the organic ligand concentration at the perturbation site. Furthermore, the optimal Ni concentration can vary considerably between phytoplankton species and as discussed above, can depend on the availability of N sources in the environment. Therefore, we need to consider not only the total inputs of Ni, but also regional differences in organic ligand concentrations, nutrient availability, and phytoplankton community composition to evaluate the potential impact of Ni on phytoplankton.

Our results suggest that excess Ni has a limited toxic impact on most of the phytoplankton species tested in our study. As the tested species cover a relatively wide range of taxa, it may be assumed that our findings can be

generalised more widely to natural communities of phytoplankton in temperate regions. However, great care must be taken when interpreting our results because we used EDTA, a strong organic ligand, in our experiments. EDTA binds large amounts of Ni so that using the total dissolved Ni concentration for inferring the absence of a toxic effect of high Ni on phytoplankton may not be valid. Although we confirmed the absence of a toxicity effect of Ni on *P. tricornutum* (CS-29) grown in Southern Ocean seawater media (which did not contain EDTA), we cannot rule out toxicity for all the other species tested here. Because *P. tricornutum* is a known "lab rat" that readily grows under a wide range of conditions, the absence of a toxicity effect in this species is not necessarily indicative of other species. Thus, we must emphasize that our results do not reject the possibility that high Ni concentrations invoked by OAE could inhibit the growth of phytoplankton.

If we assume that ligands mitigate the impacts of dissolved Ni by reducing the concentration of $Ni^{2+}$ it raises an interesting question: could such a dependency be exploited for OAE implementation strategies? As we discussed in section 4.1, open ocean ecosystems probably have lower organic ligand concentrations than many coastal or estuarine regions. Thus, under this assumption, a perturbation with Ni due to OAE would lead to a more pronounced increase in $Ni^{2+}$ in open ocean systems than the same perturbation in a coastal/estuarine region rich in organic ligands. Therefore, future research could investigate if regional differences in ligand concentration may be utilized to identify suitable spots to manage environmental impacts of OAE applications with Ni-rich minerals.

**5 Conclusion**

The Ni sensitivity of phytoplankton varied between the 11 species tested within this study but was generally rather low. This may be partly due to the use of nitrate as a nitrogen source in our experiments as other studies have revealed higher Ni sensitivities when growth is fuelled by other nitrogen-sources, such as urea. The reduced sensitivity observed in our study may also be due to the use of the high concentration of organic ligand (EDTA) added to our media, which complexed Ni making it less available for biological interactions. Considering the nitrogen sources, ligand concentration, and phytoplankton composition in test regions is important in assessing the potential environmental risks of OAE.

**Appendix A**

**Table A1.** The bioactive concentration of trace metals in different media calculated with Visual MINTEQ. Total ion concentrations of each trace metals from the original natural seawater and Aquil media were measured using seaFAST

system. The free ion concentrations were calculated based on the total ion concentrations from natural seawater or ultra-pure water together with the added concentration during the experiment. Temperature =17°C; pH 8.1; Ionic strength = 0.7.

| Metal | Aquil medium with 100 µmol/L EDTA Free ion concentration (mol/L) | no EDTA Southern Ocean seawater medium Free ion concentration (mol/L) |
|---|---|---|
| Se | $9.43 \times 10^{-9}$ | $9.43 \times 10^{-9}$ |
| Co | $7.01 \times 10^{-12}$ | $7.48 \times 10^{-11}$ |
| Zn | $1.54 \times 10^{-11}$ | $5.48 \times 10^{-10}$ |
| Cu | $1.08 \times 10^{-14}$ | $8.33 \times 10^{-11}$ |
| Mn | $2.09 \times 10^{-8}$ | $6.25 \times 10^{-9}$ |
| Fe | $5.41 \times 10^{-20}$ | $6.68 \times 10^{-19}$ |

**Table A2**. The visual MINTEQ software condition.

| pH | Ionic strength | Temperature | The calculation methods | The thermodynamic database |
|---|---|---|---|---|
| 8.1 | 0.7 | 17°C | the Davies equation | Thermo.vdb |

**Table A3.** The maximum change in growth rate, Fv/Fm and $\sigma_{PSII}$ values. The change of each treatment from each species was calculated as: $|V-V_{ave}|/V_{ave} * 100\%$ where V are treatment specific measurements and $V_{ave}$ is the average value in all 17 treatments. The maximum change is the largest change between an individual treatment and the treatment average. In species *N. closterium* (CS-5), the highest $\sigma_{PSII}$ value was 51% higher than the average values and this could be an outlier as the second highest change of *N. closterium* (CS-5) $\sigma_{PSII}$ values was 9.61%.

| Strain No. | Name | Maximum change in growth rate % | Maximum change in $F_v/F_m$ values % | Maximum change in $\sigma_{PSII}$ % |
|---|---|---|---|---|
| CS-205 | *Synechococcus* sp. | 24.48 | 35.00 | 9.03 |
| CS-897 | *Geitlerinema* sp. | 24.30 | 17.15 | 13.21 |
| CS-52 | *Oscillatoria* sp. | 37.52 | 26.67 | 10.36 |
| CS-740 | *Amphidinium carterae* | 35.98 | 11.15 | 12.02 |
| CS-1183 | *Cricosphaera* sp. | 15.12 | 7.19 | 11.17 |
| CS-1185 | *Emiliania huxleyi* | 22.91 | 5.55 | 10.24 |
| CS-186 | *Isochrysis galbana* | 15.69 | 11.10 | 8.10 |
| CS-659 | *Prymnesium parvum* | 12.64 | 6.01 | 13.30 |
| CS-135 | *Asterionellopsis glacialis* | 32.69 | 7.74 | 15.99 |
| CS-5 | *Nitzschia closterium* | 7.69 | 2.72 | 50.51 (9.61)* |
| CS-29 | *Phaeodactylum tricornutum* | 3.60 | 4.49 | 3.94 |
| CS-29SO | *Phaeodactylum tricornutum* | 9.50 | 4.61 | 15.12 |

**Author contributions**

LTB, RS, AW, and JAG designed the experiments and JAG carried them out. LTB, RS and AW supervised the study. AF and JAG conducted statistical analyses. JAG prepared the manuscript with contributions from all authors.

**Competing interests**

The authors declare that they have no conflict of interest.

**Data availability**

Data is available in Institute for Marine and Antarctic Studies (IMAS) data catalogue, University of Tasmania (UTAS). Guo, J.: Growth rate and Fast Repetition Rate fluorometry (FRRf) of phytoplankton [Data set], IMAS, https://doi.org/10.25959/1Z63-7555, 2021.

**Acknowledgements**

The authors thank Pam Quayle and Axel Durand for their assistance with the experimental infrastructure.

**Financial support**

This study was funded by the Australian Research Council by Future Fellowship FT200100846 awarded to LTB.

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
