# Peer review of "Investigating the effect of nickel concentration on phytoplankton growth to inform the assessment of ocean alkalinity enhancement"

_Biogeosciences, 2021_

## Author Response (AR1)

Reviewers comment:

Authors' response to the comment made by the two reviewers on the choice of parameter 'k' for the growth vs Ni curve fitting is not adequately responded. The authors state that the 'k-value was adjusted visually to fit the data. The choice of 'k' should be statistically sound. For instance, a minimum possible 'k' may be chosen such that the 95% confidence interval of the fit, indicated in the graph, covers the observed variability in growth rate against Ni. Or else, the allowed 'wiggle' (which increases with increasing 'k' value) in the fitted curve may be tied to the precision in the growth rate estimation.

Response:

We thank the reviewer for pointing this out and think that the following clarifications should solve any remaining doubts over the selection of the 'K' value. We first and foremost checked all statistical analyses using the function 'gam.check' in the package 'mgcv' (Wood, 2022). This check provides a number of parameters which are used to assess the selection of the 'K' value. In 'gam.check' results, the p-values are for the test of the null hypothesis that the basis dimension ('K value') used is of sufficient size. The small p-value indicates the GAM has missed a pattern or trend within the residuals (Wood, 2022). Therefore, we ensured the p-value was not 'too low' (<0.05) but please note that there is no given value which is considered to be low. If the p-value was smaller than 0.05, we increased the k value by 1 and check again by using the 'gam.check' function until the p-value was higher than 0.05. In this way, we kept the 'K' value the smallest number to be able to explain the trend of responses.

In the case of our statistical analysis, we believed that it was more important to set the 'k' value lower rather than high. In our experiment, we assume a smooth change, such as an optimal performance curve, in physiological performance in phytoplankton from different Ni concentration media. A high 'k' value resulted in smoothers that were fitted to values that in our experience would have most likely been measurement error. It is stated by Wood (2022) that the "exact choice of k is not generally critical: it should be chosen to be large enough that you are reasonably sure of having enough degrees of freedom to represent the underlying 'truth' reasonably well but small enough to maintain reasonable computational efficiency. Clearly 'large' and 'small' are dependent on the particular problem being addressed."

Reference:

Wood S., Mixed GAM computation vehicle with automatic smoothness estimation: https://cran.r-project.org/web/packages/mgcv/mgcv.pdf, last access: 20 June 2022.

---

## Author Response (AR2)

Response letter

Dear associate editor,

Thank you for your comments on my manuscript. Here is our response to these comments.

**Comments to the Author:**
In title of the manuscript - "---- to inform the assessment of ocean alkalinity enhancement" does not read well. Authors may like to consider following suggestion for title, as acceptable:

a) Investigating the effect of nickel concentration on phytoplankton growth to assess the impact of ocean alkalinity enhancement
OR
b) Investigating the effect of nickel concentration on phytoplankton growth: Implications to ocean alkalinity enhancement

**Response:**

Thank you for your excellent suggestions concerning the title. We have changed the title to "Investigating the effect of nickel concentration on phytoplankton growth to assess potential side-effects of ocean alkalinity enhancement". We think this is important to add "potential side-effects" because Nickel is not the only impact it could have. The adjustment also indicates that Ni is not something that is intentionally changed, which is important.

Cheers,

Jiaying Abby Guo